# LOCALIZING AND EDITING KNOWLEDGE IN TEXT-TO-IMAGE GENERATIVE MODELS

**Samyadeep Basu**[1], **Nanxuan Zhao**[2], **Vlad Morariu**[2], **Soheil Feizi\***[1], **Varun Manjunatha\***[2]
[1]: University of Maryland, [2]: Adobe Research;
`Correspondence to:sbasu12@umd.edu`

## ABSTRACT

Text-to-Image Diffusion Models such as Stable-Diffusion and Imagen have achieved unprecedented quality of photorealism with state-of-the-art FID scores on MS-COCO and other generation benchmarks. Given a caption, image generation requires fine-grained knowledge about attributes such as object structure, style, and viewpoint amongst others. *Where does this information reside in text-to-image generative models?* In our paper, we tackle this question and understand how knowledge corresponding to distinct visual attributes is stored in large-scale text-to-image diffusion models. We adapt Causal Mediation Analysis for text-to-image models and trace knowledge about distinct visual attributes to various (causal) components in the (i) UNet and (ii) text-encoder of the diffusion model. In particular, we show that unlike large-language models, knowledge about different attributes is not localized in isolated components, but is instead distributed amongst a set of components in the conditional UNet. These sets of components are often distinct for different visual attributes (e.g., *style / objects*). Remarkably, we find that the text-encoder in public text-to-image models such as Stable-Diffusion contains *only* one causal state across different visual attributes, and this is the first self-attention layer corresponding to the last subject token of the attribute in the caption. This is in stark contrast to the causal states in other language models which are often the mid-MLP layers. Based on this observation of *only* one causal state in the text-encoder, we introduce a fast, data-free model editing method DIFF-QUICKFIX which can effectively edit concepts (remove or update knowledge) in text-to-image models. DIFF-QUICKFIX can edit (ablate) concepts in under a second with a closed-form update, providing a significant 1000x speedup and comparable editing performance to existing fine-tuning based editing methods. Code at https://github.com/samyadeepbasu/DiffQuickFix.

## 1 INTRODUCTION

Text-to-Image generative models such as Stable-Diffusion (Rombach et al., 2021), Imagen (Saharia et al., 2022) and DALLE (Ramesh et al., 2021) have revolutionized conditional image generation in the last few years. These models have attracted a lot of attention due to their impressive image generation and editing capabilities, obtaining state-of-the-art FID scores on common generation benchmarks such as MS-COCO (Lin et al., 2014). Text-to-Image generation models are generally trained on billion-scale image-text pairs such as LAION-5B (Schuhmann et al., 2022) which typically consist of a plethora of visual concepts encompassing color, artistic styles, objects, and famous personalities, amongst others. Prior works (Carlini et al., 2023; Somepalli et al., 2023a;b) have shown that text-to-image models such as Stable-Diffusion memorize various aspects of the pre-training dataset. For example, given a caption from the LAION dataset, a model can generate an exact image from the training dataset corresponding to the caption in certain cases (Carlini et al., 2023). These observations reinforce that some form of knowledge corresponding to visual attributes is stored in the parameter space of text-to-image model.

When an image is generated, it possesses visual attributes such as (but not limited to) the presence of distinct objects with their own characteristics (such as color or texture), artistic style or scene

---

\* Equal Advising

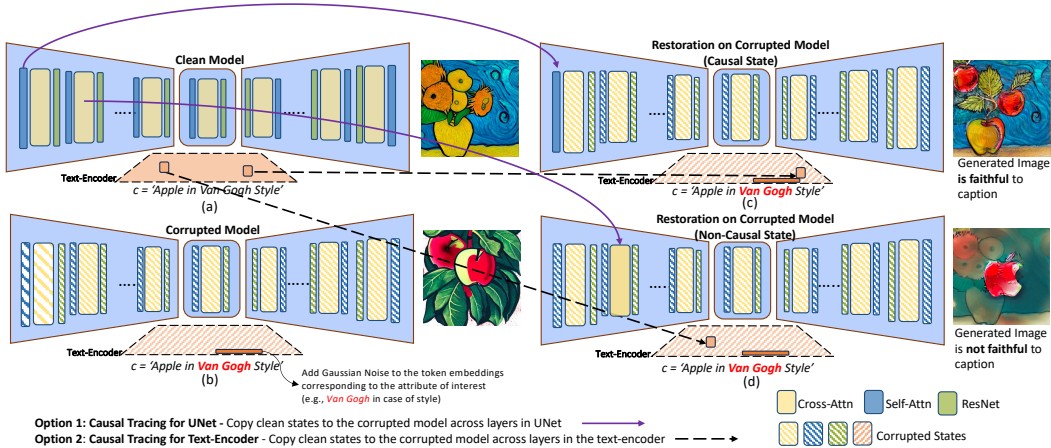

Figure 1: **Causal Tracing in Text-to-Image Models for (i) UNet and (ii) Text-Encoder shows that knowledge location matters, i.e., restoring causal layers in a corrupted model causes the model to obey the prompt again, while restoring non-causal layers does not.** (a) *Clean Model*: We prompt a Stable-Diffusion model in the conventional way and generate an image as output. (b) *Corrupted Model*: Token embeddings corresponding to attribute of interest are corrupted, leading to a generated image that does not obey the prompt. (c) *Restored (Causal) Model*: Causal layer activations are now copied from the clean model to the corrupted model. We observe that the corrupted model can now generate images with high fidelity to the original caption. (d) *Restored (Non-Causal) Model*: Non-causal layer activations are copied from the clean model to the corrupted model, but we now observe that the generated image does not obey the prompt. A single layer is copied at a time, and it can be from either the UNet (solid violet arrow) or the text-encoder (broken black arrow).

viewpoint. This attribute-specific information is usually specified in the conditioning textual prompt to the UNet in text-to-image models which is used to pull relevant knowledge from the UNet to construct and subsequently generate an image. This leads to an important question: *How and where is knowledge corresponding to various visual attributes stored in text-to-image models?* In our paper, we choose to concentrate on specific visual attributes within a scene, including objects, style, action, and color. Additionally, in the Appendix, we delve into aspects like count and viewpoint.

In this work, we empirically study this question towards understanding how knowledge corresponding to different visual attributes is stored in text-to-image models, using Stable Diffusion(Rombach et al., 2021) as a representative model. In particular, we adapt Causal Mediation Analysis (Vig et al., 2020; Pearl, 2001) for large-scale text-to-image diffusion models to identify specific causal components in the (i) UNet and (ii) the text-encoder where visual attribute knowledge resides. Previously, Causal Meditation Analysis has been used for understanding where factual knowledge is stored in LLMs. In particular, (Meng et al., 2022) find that factual knowledge is localized and stored in the mid-MLP layers of a LLM such as GPT-J (Wang & Komatsuzaki, 2021). Our work, however, paints a different picture - for multimodal text-to-image models, we specifically find that knowledge is not localized to one particular component. Instead, there exist various components in the UNet where knowledge is stored. However, each of these components store attribute information with a different efficacy and often different attributes have a distinct set of causal components where knowledge is stored. For e.g., for *style* – we find that the first self-attention layer in the UNet stores *style* related knowledge, however it is not causally important for other attributes such as *objects*, *viewpoint* or *action*. To our surprise, we specifically find that the cross-attention layers are not causally important states and a significant amount of knowledge is in fact stored in components such as the ResNet blocks and the self-attention blocks.

Remarkably, in the text-encoder, we find that knowledge corresponding to distinct attributes is strongly localized, contrary to the UNet. However unlike generative language models (Meng et al., 2022) where the mid MLP layers are causal states, we find that the first self-attention layer is causal in the CLIP based text-encoder of public text-to-image generative models (e.g., Stable-Diffusion).

Identification of local causal states in a given model has a crucial benefit: it allows for incorporating controlled edits to the model by updating *only* a tiny fraction of the model parameters without any

fine-tuning. Using our observation that the text-encoder hosts *only* one localized causal state, we introduce a new data-free and fast model editing method - DIFF-QUICKFIX which can edit concepts in text-to-image models effectively using a closed-form update. In particular, we show that DIFF-QUICKFIX can (i) remove copyrighted styles, (ii) trademarked objects as well as (iii) update stale knowledge 1000x faster than existing fine-tuning based editing methods such as (Kumari et al., 2023; Gandikota et al., 2023) with comparable or even better performance in some cases.

In summary, our contributions are as follows:

- We adapt Causal Mediation Analysis (Pearl, 2001; Vig et al., 2020) to large-scale text-to-image models (with Stable-Diffusion as a representative model), and use it to trace knowledge corresponding to various visual attributes in the UNet and text-encoder.

- We perform large-scale analysis of the identified causal components and shed light on the knowledge flow corresponding to various visual attributes in the UNet and the text-encoder.

- Leveraging the interpretability observations of localized causal states in the text-encoder, we develop a light-weight method DIFF-QUICKFIX which can edit various concepts in text-to-image models in under a second, 1000x faster than existing concept ablating methods Kumari et al. (2023); Gandikota et al. (2023).

## 2 RELATED WORKS

**Text-to-Image Diffusion Models.** In the last year, a large number of text-to-image models such as Stable-Diffusion (Rombach et al., 2021), DALLE (Ramesh et al., 2021) , Imagen (Saharia et al., 2022) and others (Balaji et al., 2022; Chang et al., 2023; Ding et al., 2022; Kang et al., 2023) have been released. In addition, the open-source community has released DeepFloyd[1] and Midjourney[2] which can generate photorealistic images given a text prompt. While most of these models operate in the latent space of the images, they differ in the text-encoder used. For e.g., Stable-Diffusion uses CLIP for the text-encoder, whereas Imagen uses T5. These text-to-image diffusion models have been used as a basis for various applications such as image-editing, semantic-segmentation, object-detection, image restoration and zero-shot classification.

**Intepretability of Text-to-Image Models.** To our knowledge, few works delve into the mechanisms of large text-to-image models like Stable-Diffusion. DAAM (Tang et al., 2023) interprets diffusion models by analyzing cross-attention maps between text tokens and images, emphasizing their semantic accuracy for interpretation. In contrast, our approach focuses on comprehending the inner workings of diffusion models by investigating the storage of visual knowledge related to different attributes. We explore various model layers beyond just the cross-attention layer.

**Editing Text-to-Image Models.** Understanding knowledge storage in diffusion models has significant implications for model editing. This ability to modify a diffusion model's behavior without retraining from scratch were first explored in Concept-Ablation (Kumari et al., 2023) and Concept-Erasure (Gandikota et al., 2023). TIME (Orgad et al., 2023) is another model editing method which translates between concepts by modifying the key and value matrices in cross-attention layers. However, the experiments in (Orgad et al., 2023) do not specifically target removing or updating concepts such as those used in (Kumari et al., 2023; Gandikota et al., 2023). We also acknowledge concurrent works (Gandikota et al., 2024) and (Anonymous, 2023) use a closed-form update on the cross-attention layers and text-encoder respectively to ablate concepts. However, we note that our work focuses primarily on first understanding how knowledge is stored in text-to-image models and subsequently using this information to design a closed-form editing method for editing concepts.

## 3 CAUSAL TRACING FOR TEXT-TO-IMAGE GENERATIVE MODELS

In this section, we first provide a brief overview of diffusion models in Sec.(3.1). We then describe how causal tracing is adapted to multimodal diffusion models such as Stable-Diffusion.

---

[1] https://www.deepfloyd.ai
[2] https://www.midjourney.com/

## 3.1 BACKGROUND

Diffusion models are inspired by non-equilibrium thermodynamics and specifically aim to learn to denoise data through a number of steps. Usually, noise is added to the data following a Markov chain across multiple time-steps $t \in [0, T]$. Starting from an initial random real image $\mathbf{x}_0$, the noisy image at time-step $t$ is defined as $\mathbf{x}_t = \sqrt{\alpha_t} \mathbf{x}_0 + \sqrt{(1 - \alpha_t)} \epsilon$. In particular, $\alpha_t$ determines the strength of the random Gaussian noise and it gradually decreases as the time-step increases such that $\mathbf{x}_T \sim \mathcal{N}(0, I)$. The denoising network denoted by $\epsilon_\theta(\mathbf{x}_t, \mathbf{c}, t)$ is pre-trained to denoise the noisy image $\mathbf{x}_t$ to obtain $\mathbf{x}_{t-1}$. Usually, the conditional input $\mathbf{c}$ to the denoising network $\epsilon_\theta(.)$ is a text-embedding of a caption $c$ through a text-encoder $\mathbf{c} = v_\gamma(c)$ which is paired with the original real image $\mathbf{x}_0$. The pre-training objective for diffusion models can be defined as follows for a given image-text pair denoted by $(\mathbf{x}, \mathbf{c})$:

$$\mathcal{L}(\mathbf{x}, \mathbf{c}) = \mathbb{E}_{\epsilon, t} ||\epsilon - \epsilon_\theta(\mathbf{x}_t, \mathbf{c}, t)||_2^2, \tag{1}$$

where $\theta$ is the set of learnable parameters. For better training efficiency, the noising as well as the denoising operation occurs in a latent space defined by $\mathbf{z} = \mathcal{E}(\mathbf{x})$ Rombach et al. (2021). In this case, the pre-training objective learns to denoise in the latent space as denoted by:

$$\mathcal{L}(\mathbf{x}, \mathbf{c}) = \mathbb{E}_{\epsilon, t} ||\epsilon - \epsilon_\theta(\mathbf{z}_t, \mathbf{c}, t)||_2^2, \tag{2}$$

where $\mathbf{z}_t = \mathcal{E}(\mathbf{x}_t)$ and $\mathcal{E}$ is an encoder such as VQ-VAE (van den Oord et al., 2017). During inference, where the objective is to synthesize an image given a text-condition $\mathbf{c}$, a random Gaussian noise $\mathbf{x}_T \sim \mathcal{N}(0, I)$ is iteratively denoised for a fixed range of time-steps in order to produce the final image. We provide more details on the pre-training and inference steps in Appendix L.

## 3.2 ADAPTING CAUSAL TRACING FOR TEXT-TO-IMAGE DIFFUSION MODELS

Causal Mediation Analysis (Pearl, 2001; Vig et al., 2020) is a method from causal inference that studies the change in a response variable following an intervention on intermediate variables of interest (mediators). One can think of the internal model components (e.g., specific neurons or layer activations) as mediators along a directed acyclic graph between the input and output. For text-to-image diffusion models, we use Causal Mediation Analysis to trace the causal effects of these internal model components within the UNet and the text-encoder which contributes towards the generation of images with specific visual attributes (e.g., *objects, style*). For example, we find the subset of model components in the text-to-image model which are causal for generating images with specific *objects*, *styles*, *viewpoints*, *action* or *color*.

**Where is Causal Tracing Performed?** We identify the causal model components in both the UNet $\epsilon_\theta$ and the text-encoder $v_\gamma$. For $\epsilon_\theta$, we perform the causal tracing at the granularity of layers, whereas for the text-encoder, causal tracing is performed at the granularity of hidden states of the token embeddings in $\mathbf{c}$ across distinct layers. The UNet $\epsilon_\theta$ consists of 70 unique layers distributed amongst three types of blocks: (i) `down-block`; (ii) `mid-block` and (iii) `up-block`. Each of these blocks contain varying number of cross-attention layers, self-attention layers and residual layers. Fig 1 visualizes the internal states of the UNet and how causal tracing for knowledge attribution is performed. For the text-encoder $v_\gamma$, there are 12 blocks in total with each block consisting of a self-attention layer and a MLP layer (see Fig 1). We highlight that the text-encoder in text-to-image models such as Stable-Diffusion has a GPT-style architecture with a causal self-attention, though it's pre-trained without a language modeling objective. More details on the layers used in Appendix J.

Given a caption $c$, an image $\mathbf{x}$ is generated starting from some random Gaussian noise. This image $\mathbf{x}$ encapsulates the visual properties embedded in the caption $c$. For e.g., the caption $c$ can contain information corresponding from *objects* to *action* etc. We specifically identify distinct components in the UNet and the text-encoder which are causally responsible for these properties.

**Creating the Probe Captions.** We primarily focus on four different visual attributes for causal tracing: (i) *objects*; (ii) *style*; (iii) *color*; and (iv) *action*. In particular, identifying the location of knowledge storage for *objects* and *style* can be useful to perform post-hoc editing of diffusion models to edit concepts (e.g., delete or update certain concepts). We provide the complete details about the probe dataset used for causal tracing in Appendix A. The probe dataset also contains additional captions for *viewpoint* and *count* attribute. However, we do not focus on them as often the generations from the unedited model are erroneous for these attributes (see Appendix E for details).

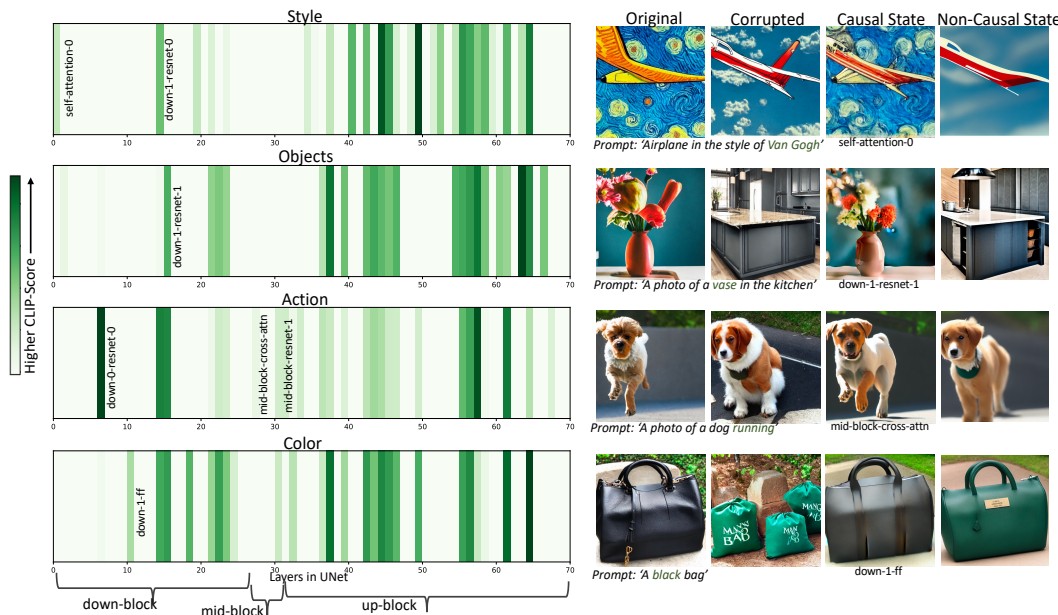

Figure 2: **Causal Tracing Results for the UNet: Knowledge is Distributed.** The intensity of the bars indicate the CLIP-Score between the generated image (after causal intervention) and the original caption. For each attribute, we find that the causal states are distributed across the UNet and the distribution varies amongst distinct attributes. For e.g., self-attn in the first layer is causal for *style*, but not for *objects*, *action* or *color*. Similarly, mid-block cross-attn is causal for *action*, but not for the other attributes. On the right-side, we visualize the images generated by (i) Original model; (ii) Corrupted Model; (iii) Restored causal states and (iv) Restored non-causal states in the UNet for *style, action, object, color* attributes.

## 3.3 TRACING KNOWLEDGE IN UNET

During inference, classifier-free guidance (Ho & Salimans, 2021) is used to regulate image-generation by incorporating scores from the conditional and unconditional diffusion model at each of the time-steps. In particular, at each time-step, classifier-free guidance is used in the following way to combine the conditional ($\epsilon_\theta(\mathbf{z}_t, \mathbf{c}, t)$) and unconditional score estimates ($\epsilon_\theta(\mathbf{z}_t, t)$) at each time-step $t$ to obtain the combined score denoted as $\hat{\epsilon}(\mathbf{z}_t, \mathbf{c}, t)$:

$$\hat{\epsilon}_\theta(\mathbf{z_t}, \mathbf{c}, t) = \epsilon_\theta(\mathbf{z_t}, \mathbf{c}, t) + \alpha(\epsilon_\theta(\mathbf{z_t}, \mathbf{c}, t) - \epsilon_\theta(\mathbf{z_t}, t)), \quad \forall t \in [T, 1]. \quad (3)$$

This combined score is used to update the latent $\mathbf{z}_t$ using DDIM sampling (Song et al., 2020) at each time-step iteratively to obtain the final latent code $\mathbf{z}_0$.

To perform causal tracing on the UNet $\epsilon_\theta$ (see Fig 1 for visualization), we perform a sequence of operations that is somewhat analogous to earlier work from (Meng et al., 2022) which investigated knowledge-tracing in large language models. We consider three types of model configurations: (i) a clean model $\epsilon_\theta$, where classifier-free guidance is used as default; (ii) a corrupted model $\epsilon_\theta^{corr}$, where the word embedding of the subject (e.g., *Van Gogh*) of a given attribute (e.g., *style*) corresponding to a caption $c$ is corrupted with Gaussian Noise; and, (iii) a restored model $\epsilon_\theta^{restored}$, which is similar to $\epsilon_\theta^{corr}$ except that one of its layers is restored from the clean model at each time-step of the classifier-free guidance. Given a list of layers $\mathcal{A}$, let $a_i \in \mathcal{A}$ denote the $i^{th}$ layer whose importance needs to be evaluated. Let $\epsilon_\theta[a_i]$, $\epsilon_\theta^{corr}[a_i]$ and $\epsilon_\theta^{restored}[a_i]$ denote the activations of layer $a_i$. To find the importance of layer $a_i$ for a particular attribute embedded in a caption $c$, we perform the following replacement operation on the corrupted model $\epsilon_\theta^{corr}$ to obtain the restored model $\epsilon_\theta^{restored}$:

$$\epsilon_\theta^{restored}[a_i] : \epsilon_\theta^{corr}[a_i] = \epsilon_\theta[a_i]. \quad (4)$$

Next, we obtain the restored model by replacing the activations of layer $a_i$ of the corrupted model with those of the clean model to get a restored layer $\epsilon_\theta^{restored}[a_i]$. We run classifier-free guidance to obtain the combined score estimate:

$$\hat{\epsilon}_\theta^{restored}(\mathbf{z_t}, \mathbf{c}, t) = \epsilon_\theta^{restored}(\mathbf{z_t}, \mathbf{c}, t) + \alpha(\epsilon_\theta^{restored}(\mathbf{z_t}, \mathbf{c}, t) - \epsilon_\theta^{restored}(\mathbf{z_t}, t)), \quad \forall t \in [T, 1]. \quad (5)$$

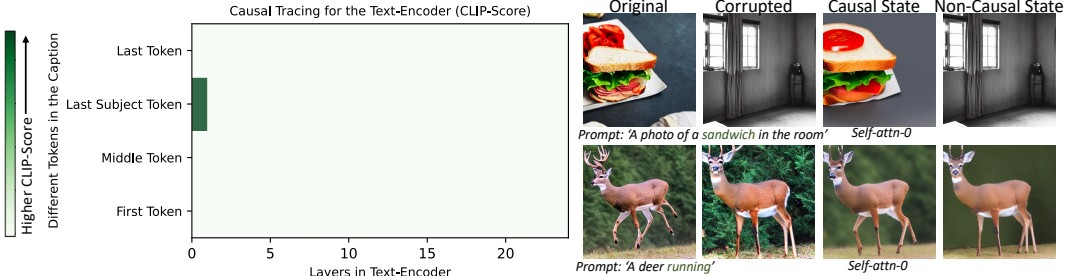

Figure 3: **Causal Tracing in the Text-Encoder: Knowledge is Localized.** In the CLIP text-encoder used for Stable-Diffusion, we find the existence of *only* one causal state, which is the first self-attention layer corresponding to the last subject token. The CLIP-Score(Left) is computed across all the four visual attributes. Visualizations (Right) further illustrate that restoring the sole causal state (self-attn-0) leads to image generation with high fidelity to the original captions.

The final latent $\mathbf{z}_0$ is obtained with the score from Equation (5) at each time-step using DDIM (Song et al., 2020) and passed through the VQ-VAE decoder to obtain the final image $\mathbf{x}_0^{restored}$.

## 3.4 TRACING KNOWLEDGE IN THE TEXT-ENCODER

The text-encoder in public text-to-image models such as Stable-Diffusion is a CLIP-ViT-L/336px text-encoder Rombach et al. (2021). Similar to Sec.(3.3), we define three states of the CLIP text-encoder: (i) Clean model denoted by $v_\gamma$; (ii) Corrupted model $v_\gamma^{corr}$ where the word embedding of the subject in a given caption $c$ is corrupted; (iii) Restored model $v_\gamma^{restored}$ which is similar to $v_\gamma^{corr}$ except that one of its layers is copied from $v_\gamma$. Similar to Sec.(3.3), to find the effect of the layer $a_i \in \mathcal{A}$, where $\mathcal{A}$ consists of all the layers to probe in the CLIP text-encoder:

$$v_\gamma^{restored}[a_i] : v_\gamma^{corr}[a_i] = v_\gamma[a_i], \tag{6}$$

We then use the restored text-encoder $v_\gamma^{restored}$ with classifier-free guidance to obtain the final score estimate:

$$\hat{\epsilon}_\theta(\mathbf{z_t}, \mathbf{c}', t) = \epsilon_\theta(\mathbf{z_t}, \mathbf{c}', t) + \alpha(\epsilon_\theta(\mathbf{z_t}, \mathbf{c}', t) - \epsilon_\theta(\mathbf{z_t}, t)), \quad \forall t \in [T, 1] \tag{7}$$

where $\mathbf{c}' = v_\gamma^{restored}[a_i](c)$ for a given caption $c$. This score estimate $\hat{\epsilon}_\theta(\mathbf{z_t}, \mathbf{c}', t)$ at each time-step $t$ is used to obtain the final latent code $\mathbf{z}_0$ which is then used with the VQ-VAE decoder to obtain the final image $\mathbf{x}_0^{restored}$.

## 3.5 EXTRACTING CAUSAL STATES USING CLIP-SCORE

In this section, we discuss details on how to retrieve causal states using automated metrics such as CLIP-Score (Hessel et al., 2021). Let $\mathbf{x}_0^{restored}(a_i)$ be the final image generated by the diffusion model after intervening on layer $a_i$, $\mathbf{x}_0$ be the image generated by the clean diffusion model and $\mathbf{x}^{corr}$ be the final image generated by the corrupted model. In particular, we are interested in the average indirect effect (Vig et al., 2020; Pearl, 2001) which measures the difference between the corrupted model and the restored model. Intuitively, a higher value of average indirect effect (AIE) signifies that the restored model deviates from the corrupted model. To compute the average indirect effect with respect to causal mediation analysis for text-to-image models such as Stable-Diffusion, we use CLIP-Score which computes the similarity between an image embedding and a caption embedding. In particular, AIE = $(\text{CLIPScore}(\mathbf{x}_0^{restored}, c) - \text{CLIPScore}(\mathbf{x}_0^{corr}, c))$. Given $\mathbf{x}_0^{corr}$ is common across all the layers, we can use $\text{CLIPScore}(\mathbf{x}_0^{restored}, c)$ as the AIE.

**Selecting Threshold for CLIP-Score.** We observe that the difference between the CLIP-Score of generated images (after restoration) with high fidelity to the original caption and generated images (after restoration) with low fidelity to the original caption, to be small. Therefore to effectively find a reasonable cut-off point to automatically select causal states (where the generated images have high-fidelity to the original caption), we use a threshold selection mechanism. In order to determine the optimal threshold value for CLIP-Score, we select a small validation set of 10 prompts per attribute. To this end, we establish a concise user study interface (refer to Appendix D for details).

Through human participation, we collect binary ratings if an image generated by restoring a particular layer is faithful to the original captions. We then extract the common causal states across all the prompts for a given attribute and find the average (across all the prompts) `CLIP-Score` for each causal state. We then use the lowest average `CLIP-Score` corresponding to a causal state as the threshold, which we apply on the probe dataset in Appendix A to filter the causal states at scale for each attribute separately.

## 4   HOW IS KNOWLEDGE STORED IN TEXT-TO-IMAGE MODELS?

In this section, we discuss the results of tracing knowledge across various components of the text-to-image model in details.

**Tracing Results for UNet.** In Fig 2, we illustrate the distribution of causal states across different visual attributes within the UNet architecture using the `CLIP-Score` metric. This metric evaluates the faithfulness of the image produced by the restored state $\mathbf{x}_0^{restored}$ compared to the original caption $c$. From the insights derived in Fig 2, it becomes evident that causal states are spread across diverse components of the UNet. In particular, we find that the density of the causal states are more in the `up-block` of the UNet when compared to the `down-block` or the `mid-block`. Nonetheless, a notable distinction emerges in this distribution across distinct attributes. For instance, when examining the *style* attribute, the initial self-attention layer demonstrates causality, whereas this causal relationship is absent for other attributes. Similarly, in the context of the *action* attribute, the cross-attention layer within the mid-block exhibits causality, which contrasts with its non-causal behavior concerning other visual attributes. Fig 2 showcases the images generated by restoring both causal and non-causal layers within the UNet. A comprehensive qualitative enumeration of both causal and non-causal layers for each visual attribute is provided in Appendix B. Our findings underscore the presence of information pertaining to various visual attributes in regions beyond the cross-attention layers. Importantly, we observe that the distribution of information within the UNet diverges from the patterns identified in extensive generative language models, as noted in prior research (Meng et al., 2022), where attribute-related knowledge is confined to a few proximate layers. In Appendix M, we provide additional causal tracing results, where we add Gaussian noise to the entire text-embedding. Even in such a case, certain causal states can restore the model close to its original configuration, highlighting that the conditional information can be completely bypassed if certain causal states are active.

**Tracing Results for Text-Encoder.** In Fig 3, we illustrate the causal states in the text-encoder for Stable-Diffusion corresponding to various visual attributes. At the text-encoder level, we find that the causal states are localized to the first self-attention layer corresponding to the last subject token across all the attributes. In fact, there exists *only one* causal state in the text-encoder. Qualitative visualizations in Fig 3 and Appendix C illustrate that the restoration of layers other than the first self-attention layer corresponding to the subject token does not lead to images with high fidelity to the original caption. Remarkably, this observation is distinct from generative language models where factual knowledge is primarily localized in the proximate mid MLP layers Meng et al. (2022).

> **General Takeaway.** Causal components corresponding to various visual attributes are dispersed (with a *different distribution* between *distinct* attributes) in the UNet, whereas there exists *only* one causal component in the text-encoder.

The text-encoder's strong localization of causal states for visual attributes enables controlled knowledge manipulation in text-to-image models, facilitating updates or removal of concepts. However, since attribute knowledge is dispersed in the UNet, targeted editing is challenging without layer interference. While fine-tuning methods for UNet model editing exist (Gandikota et al., 2023; Kumari et al., 2023), they lack scalability and don't support simultaneous editing of multiple concepts. In the next section, we introduce a closed-form editing method, DIFF-QUICKFIX, leveraging our causal tracing insights to efficiently edit various concepts in text-to-image models.

## 5   DIFF-QUICKFIX: FAST MODEL EDITING FOR TEXT-TO-IMAGE MODELS

### 5.1   EDITING METHOD

Recent works such as (Kumari et al., 2023; Gandikota et al., 2023) edit concepts from text-to-image diffusion models by fine-tuning the UNet. They generate training data for fine-tuning using the pretrained diffusion model itself. While both methods are effective at editing concepts, fine-tuning the

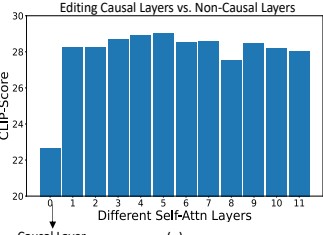 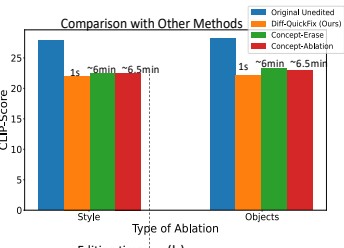 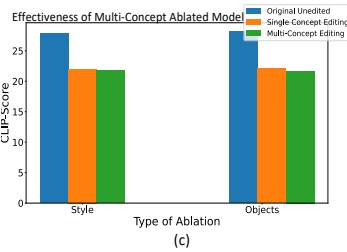

Figure 4: **Quantitative Analysis of DIFF-QUICKFIX**. (a) Editing Causal vs. Non-Causal Layers (Averaged across *Objects, Style and Facts*): Lower `CLIP-Score` for causal layer indicates successful edits; (b) Efficacy of DIFF-QUICKFIX when compared to other methods – Our method leads to comparable `CLIP-Scores` to fine-tuning based approaches, but can edit concepts 1000x faster; (c) DIFF-QUICKFIX can be used to effectively edit multiple concepts at once, shown by comparable `CLIP-Scores` to the single-concept edited ones.

UNet can be expensive due to backpropagation of gradients through the UNet. To circumvent this issue, we design a fast, data-free model editing method leveraging our interpretability observations in Section 4, where we find that there exists only one causal state (the very first self-attention layer) in the text-encoder for Stable-Diffusion.

Our editing method DIFF-QUICKFIX can update text-to-image diffusion models in a targeted way in under $1s$ through a closed-form update making it 1000x faster than existing fine-tuning based concept ablating methods such as (Kumari et al., 2023; Gandikota et al., 2023). The first self-attention layer in the text-encoder for Stable-Diffusion contains four updatable weight matrices: $W_k, W_q, W_v$ and $W_{out}$, where $W_k, W_q, W_v$ are the projection matrices for the key, query and value embeddings respectively. $W_{out}$ is the projection matrix before the output from the `self-attn-0` layer after the attention operations. DIFF-QUICKFIX specifically updates this $W_{out}$ matrix by collecting caption pairs $(c_k, c_v)$ where $c_k$ (key) is the original caption and $c_v$ (value) is the caption to which $c_k$ is mapped. For e.g., to remove the style of *'Van Gogh'*, we set $c_k = $ *'Van Gogh'* and $c_v = $ *'Painting'*. In particular, to update $W_{out}$, we solve the following optimization problem:

$$\min_{W_{out}} \sum_{i=1}^{N} \|W_{out}k_i - v_i\|_2^2 + \lambda\|W_{out} - W'_{out}\|_2^2, \qquad (8)$$

where $\lambda$ is a regularizer to not deviate significantly from the original pre-trained weights $W'_{out}$, $N$ denotes the total number of caption pairs containing the last subject token embeddings of the key and value. $k_i$ corresponds to the embedding of $c_{k_i}$ after the attention operation using $W_q, W_k$ and $W_v$ for the $i^{th}$ caption pair. $v_i$ corresponds to the embedding of $c_{v_i}$ after the original pre-trained weights $W'_{out}$ acts on it.

One can observe that Eq. (8) has a closed-form solution due to the absence of any non-linearities. In particular, the optimal $W_{out}$ can be expressed as the following:

$$W_{out} = (\lambda W'_{out} + \sum_{i=1}^{N} v_i k_i^T)(\lambda I + \sum_{i=1}^{N} k_i k_i^T)^{-1}, \qquad (9)$$

In Section 5.3, we show qualitative as well as quantitative results using DIFF-QUICKFIX for editing various concepts in text-to-image models.

## 5.2 EXPERIMENTAL SETUP

We validate DIFF-QUICKFIX by applying edits to a Stable-Diffusion (Rombach et al., 2021) model and quantifying the *efficacy* of the edit. For removing concepts such as artistic styles or objects using DIFF-QUICKFIX, we use the prompt dataset from (Kumari et al., 2023). For updating knowledge (e.g., *President of a country*) in text-to-image models, we add newer prompts to the prompt dataset from (Kumari et al., 2023) and provide further details in Appendix N. We compare our method with (i) Original Stable-Diffusion; (ii) Editing methods from (Kumari et al., 2023) and (Gandikota et al., 2023). To validate the effectiveness of editing methods including our DIFF-QUICKFIX, we perform evaluation using automated metrics such as `CLIP-Score`. In particular, we compute the `CLIP-Score` between the images from the edited model and the concept corresponding to the visual attribute which is edited. A low `CLIP-Score` therefore indicates correct edits.

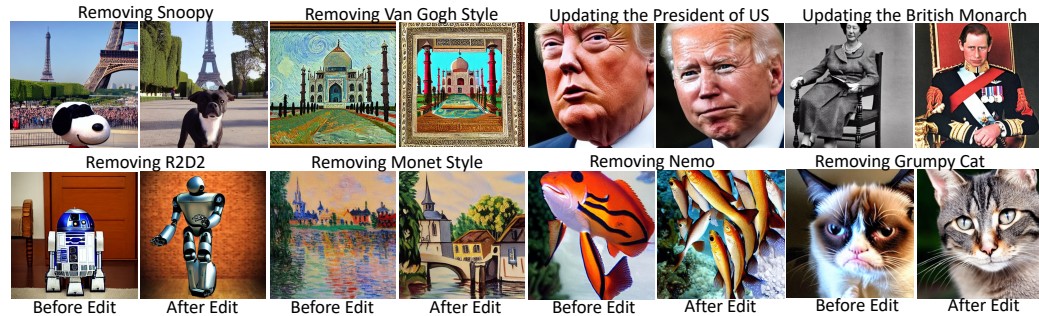

Figure 5: **Qualitative Examples with using DIFF-QUICKFIX** to ablate *style*, *objects* and update *facts* in text-to-image models. More qualitative examples in the Appendix F.

## 5.3 EDITING RESULTS

**Editing Non-causal Layers Does Not Lead to Correct Edits.** We use DIFF-QUICKFIX with the non-causal self-attention layers in the text-encoder to ablate *styles*, *objects* and update *facts*. In Fig 4-(a), we compute the CLIP-Score between the generated images and the attribute from the original captions (e.g., *van gogh* in the case of *style*). In particular, we find that editing the non-causal layers does not lead to any intended model changes – highlighted by the high CLIP-Scores consistently across non-causal layers (layers numbered 1 to 11). However, editing the sole causal layer (layer-0) leads to correct model changes, highlighted by the lower CLIP-Score between the generated images from the edited model and the attribute from the original captions. This shows that identifying the causal states in the model is particularly important to perform targeted model editing for ablating concepts. In Appendix G, we show additional qualitative visualizations.

**Efficacy in Removing Styles and Objects**. Fig 4-(b) shows the average CLIP-Score of the generated images from the edited model computed with the relevant attributes from the original captions. We find that the CLIP-Score from the edited model with DIFF-QUICKFIX decreases when compared to the generations from the unedited model. We also find that our editing method has comparable CLIP-Scores to other fine-tuning based approaches such as Concept-Erase (Gandikota et al., 2023) and Concept-Ablation (Kumari et al., 2023), which are more computationally expensive. Fig 5 shows qualitative visualizations corresponding to images generated by the text-to-image model before and after the edit operations. Together, these quantitative and qualitative results show that DIFF-QUICKFIX is able to effectively remove various *styles* and *objects* from an underlying text-to-image model. In Appendix F we provide additional qualitative visualizations and in Fig 52 we show additional results showing that our editing method does not harm surrounding concepts (For e.g., removing the style of *Van Gogh* does not harm the style of *Monet*).

**Efficacy in Updating Stale Knowledge**. The CLIP-Score between the generated images and a caption designating the incorrect fact (e.g., *Donald Trump* as the *President of the US*) decreases from 0.28 to 0.23 after editing with DIFF-QUICKFIX, while the CLIP-Score with the correct fact (e.g., *Joe Biden* as the *President of the US*) increases from 0.22 to 0.29 after the relevant edit. This shows that the incorrect fact is updated with the correct fact in the text-to-image model. Additional qualitative visualizations are provided in Fig 5 and Appendix F.

**Multiple Edits using DIFF-QUICKFIX**. An important feature of DIFF-QUICKFIX is its capability to ablate multiple concepts simultaneously. In Fig 4-(c), our framework demonstrates the removal of up to 10 distinct styles and objects at once. This multi-concept ablation results in lower CLIP-Scores compared to the original model, similar CLIP-Scores to single concept editing.

## 6 CONCLUSION

Through the lens of Causal Mediation Analysis, we present methods for understanding the storage of knowledge corresponding to visual attributes in text-to-image models. Notably, we find a distinct distribution of causal states across attributes in the UNet, while the text-encoder maintains a single causal state. This differs significantly from observations in language models like GPT, where factual information is concentrated in mid-MLP layers. In contrast, our analysis shows that public text-to-image models like Stable-Diffusion concentrate multiple visual attributes within the first self-attention layer of the text-encoder. Harnessing these insights, we design a fast model editing method DIFF-QUICKFIX. The potency of DIFF-QUICKFIX is manifested through its adeptness in removing artistic styles, objects, and updating outdated knowledge all accomplished data-free and in less than a second, making DIFF-QUICKFIX a practical asset for real-world model editing scenarios.

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
