# OpenReview forum: "Localizing and Editing Knowledge In Text-to-Image Generative Models"
_ICLR.cc/2024/Conference — ICLR 2024 poster_

### Official Review · Reviewer_cqRW · 2023-10-31

**Soundness:** 3 good
**Presentation:** 4 excellent
**Contribution:** 3 good
**Rating:** 8
**Confidence:** 3

**Summary:**

This paper explores how knowledge corresponding to distinct visual attributes is stored in large-scale text-to-image diffusion models. The authors apply Causal Mediation Analysis to trace the various attributes in U-Net and text-encoder of diffusion models. In addition, the authors introduces an editing method, DIFF-QUICKFIX to edit the image based on their proposed method. DIFF-QUICKFIX can remove or update concepts in images.

**Strengths:**

Overview, the paper is well-written and the motivation is clear. Extensive experiments are performed to demonstrate the meanings of exploring how the attribute information is reflected in the embeddings of text-to-image generative models.
Though I'm not an expert in causal mediation analysis, I enjoy reading the paper and be happy with the abundant visualization results in the paper.

**Weaknesses:**

It would be nice if the authors can include more discussions on the limitations of the DIFF-QUICKFIX parts. *E.g.*, what can DIFF-QUICKFIX do in editing tasks. When I check the **remove** results generated by DIFF-QUICKFIX, such as "Snoppy" in Figure 5, it seems the "Snoppy" is replaced by a dog rather than removed from the figure. From my view, it is more like attribute editing rather than object removal. Is there any potential methods that can leverage DIFF-QUICKFIX to achieve various editing tasks such as remove object, add object rather than simply editing the attributes of existing objects?

In addition, it would be nice if the authors can add some descriptions in Figure captions. *E.g.*, in Figure 6, I'm confused between "photo of a apple in a beach" and "photo of a apple in a city". Are there any clues that can help readers to understand why one figure corresponds to "beach" and the other corresponds to "city"? Maybe add descriptions just like Figure 5 would be helpful.

**Questions:**

See weaknesses

---

> ### Author Response · Authors · 2023-11-18
> **Response to Reviewer**
>
> We thank the reviewer for the constructive comments and appreciating our experiments and findings!
> Below we answer the questions:
>
> **It would be nice if the authors can include more discussions on the limitations of the DIFF-QUICKFIX parts. E.g., what can DIFF-QUICKFIX do in editing tasks. When I check the remove results generated by DIFF-QUICKFIX, such as "Snoppy" in Figure 5, it seems the "Snoppy" is replaced by a dog rather than removed from the figure. From my view, it is more like attribute editing rather than object removal. Is there any potential methods that can leverage DIFF-QUICKFIX to achieve various editing tasks such as remove object, add object rather than simply editing the attributes of existing objects?** : The reviewer raises an interesting point!
> We highlight that our work follows similar evaluation protocols as [1] and use the same definition of concept-removal as theirs.
> In principle, our framework can also be used to remove an object (e.g., snoopy) completely, by mapping the object (e.g., ‘snoopy’) to a keyword which does not contain any attribute information (e.g., ‘a’, ‘the’). In Sec.(V) - Appendix, we have added new results where we show the effectiveness of our framework to also remove objects as a whole from the underlying image. The choice of the edit string could indeed be crucial and needs to be studied in more detail!
>
>
> [1]. Nupur Kumari, Bingliang Zhang, Sheng-Yu Wang, Eli Shechtman, Richard Zhang, and Jun-Yan Zhu. Ablating concepts in text-to-image diffusion models. In ’International Conference on Computer Vision (ICCV)’, 2023.
>
> **In addition, it would be nice if the authors can add some descriptions in Figure captions. E.g., in Figure 6, I'm confused between "photo of an apple in a beach" and "photo of an apple in a city". Are there any clues that can help readers to understand why one figure corresponds to "beach" and the other corresponds to "city"? Maybe add descriptions just like Figure 5 would be helpful.** : We have added more descriptions in  Fig (6) to be better readable!
> In particular, Fig. (6) shows that restoring the down-1-resnet-1 layer in the corrupted text-to-image model is able to retrieve the correct object in the generations for the majority of the captions..  However, for both these captions (as asked by the reviewer) we find that the background does not succinctly appear in the original generations. Therefore even if the restoration leads to the correct object generations, the backgrounds are not succinctly generated. We provide these original generations corresponding to all the captions  in Sec. (T) - Appendix.

---

> > ### Author Response · Authors · 2023-11-21
> > **Looking Forward To Your Response!**
> >
> > We thank you again for your valuable feedback and comments which has helped to strengthen our paper. As the discussion period is ending soon, we would really appreciate if you could let us know if our responses have addressed your concerns. We will be happy to answer any further questions and address any remaining concerns to the best of our abilities in the remaining time!

---

### Official Review · Reviewer_cjqU · 2023-11-01

**Soundness:** 2 fair
**Presentation:** 3 good
**Contribution:** 2 fair
**Rating:** 5
**Confidence:** 3

**Summary:**

The paper studies the source of knowledge about scenes being generated within the architectures of text-to-image diffusion models. It draws the conclusion that such knowledge is distributed within the conditional UNet. It further introduces an image editing method that provides speedup compared to prior arts.

**Strengths:**

* The paper adapts Causal Mediation Analysis to interpret text-to-image diffusion models and draw several meaningful conclusions in terms of the location of visual knowledge within these models.
* The paper proposes an editing method based on the observations as an application that achieves empirical advantages compared to prior methods.

**Weaknesses:**

* The main toolbox of the interpretation method, Causal Mediation Analysis, is borrowed from previous works. There is a limited novelty in terms of the interpretation framework.
* The experiments presented in the paper all use Stable-Diffusion. The results would be more convincing if other classes of diffusion models could be investigated, which would provide important cues on whether the observations are specific to the Stable-Diffusion architecture or can be transferred to other models that adopt diffusion-based training.
* The causal states are retrieved solely relying on CLIP-Score, but it's possible that such a score is sensitive to only some prominent visual attributes such as colors. The paper claims to identify the knowledge source of general visual attributes but does not provide an investigation on the efficacy of CLIP-Score to discriminate different kinds of visual attributes.

**Questions:**

* The paper highlights the difference in causal state locations between Stable-Diffusion and GPT. Is such a difference specific to the CLIP encoder or is general to the diffusion model class?

---

> ### Author Response · Authors · 2023-11-18
> **Response to Reviewer - Part 1**
>
> We thank the reviewer for the constructive comments and in appreciating our findings!
> Below, we answer the reviewer's questions:
>
> **The main toolbox of the interpretation method, Causal Mediation Analysis, is borrowed from previous works. There is a limited novelty in terms of the interpretation framework.** : We highlight that Causal Mediation Analysis [1] is a tool from the causality literature which has been used in well-cited and impactful papers ([2]). While we borrow causal mediation analysis from the existing causality framework, to the best of our knowledge we are the first to use it to understand the inner workings of text-to-image models in detail.
> We also emphasize that it is not trivial to adapt causal mediation analysis to diffusion models (especially large text-to-image models such as Stable-Diffusion). There are differences in not only architecture (e.g., the UNet in SD has several modules such as self-attention, cross-attention, residual blocks at various granularities), but also in sampling objectives where diffusion models have an added complexity of sampling over multiple time-steps.
> Moreover, our adaptation of causal mediation analysis for text-to-image models has revealed many interesting findings for e.g., Knowledge is distributed in the UNet which makes it difficult to edit concepts in the UNet, while it is localized in the text-encoder which makes it easier to perform closed-form edits.  These are significantly novel scientific findings and can help improve the understanding of large text-to-image models.
> We believe that using well-accepted and published tools from the community to reveal new scientific findings is a more constructive way to perform research, as it leads to greater adoption of existing research works by the community. Had we introduced an altogether different toolbox, the reviewer would need to evaluate this new toolbox in its entirety, in addition to our other findings. We sincerely hope that the reviewer appreciates this perspective!
>
> [1]. Judea Pearl. Direct and indirect effects. In Proceedings of the Seventeenth Conference on Uncertainty in Artificial Intelligence, UAI’01, pp. 411–420, San Francisco, CA, USA, 2001. Morgan Kaufmann Publishers Inc. ISBN 1558608001
>
> [2]. Kevin Meng, David Bau, Alex Andonian, and Yonatan Belinkov. Locating and editing factual associations in GPT. Advances in Neural Information Processing Systems, 36, 2022
>
> **The experiments presented in the paper all use Stable-Diffusion. The results would be more convincing if other classes of diffusion models could be investigated, which would provide important cues on whether the observations are specific to the Stable-Diffusion architecture or can be transferred to other models that adopt diffusion-based training.** : In our updated version of the paper, we have added new experiments using our framework on other Stable-Diffusion versions (v2-1, v1-5), OpenJourney, Kandinsky and SD-XL architectures.  Our current results are consistent across the newer SD versions too. Across all these models, we find that knowledge is distributed in the UNet (see Sec. R) - Appendix), whereas it is localized in the text-encoder (see Sec.Q). Also we find that our method can effectively edit concepts in all these models (Sec Q.1 - Appendix and Sec. Q.2, Sec. Q.2.1).
>
> For SD-XL[1] (see Sec. (S)) which has a slightly different architecture than Stable-Diffusion, we have the following observations : In SD-XL, which is a recent text-to-image model consisting of a much larger UNet (consisting of 2.6B parameters) – we find that causal states are distributed like the Stable-Diffusion checkpoints. However for SD-XL we find that only 12 causal layers exist out of 227 layers (~5% of the total layers).  This shows that our interpretability framework can be effectively used to probe other text-to-image generative models. As the Reviewer TjyF noted that for each of the models, laborious experiments are needed – a full study of SD-XL is attributed for future work.  In Sec. (R.2) - we show successful edits on OpenJourney[2].
>
> In Kandinsky[3] text-to-image generative model (see Sec. (U)) – we find only one localized causal state in the text-encoder, thereby enabling fast edits using DiffQuickFix. Therefore each component of our paper can be used first to probe any text-to-image models and then edit them if localized causal states exist.
>
> [1].Podell et al. 2023, arXiv. https://arxiv.org/pdf/2307.01952.pdf
>
> [2]. OpenJourney: https://huggingface.co/prompthero/openjourney
>
> [3]. Kandinsky: https://huggingface.co/docs/diffusers/api/pipelines/kandinsky

---

> > ### Author Response · Authors · 2023-11-18
> > **Response to Reviewer - Part 2**
> >
> > **The causal states are retrieved solely relying on CLIP-Score, but it's possible that such a score is sensitive to only some prominent visual attributes such as colors. The paper claims to identify the knowledge source of general visual attributes but does not provide an investigation on the efficacy of CLIP-Score to discriminate different kinds of visual attributes.** : While CLIP-Score is not the perfect metric it is still widely accepted by the community and still used for evaluating text-to-image generative models. For the attributes which we study in our paper, we find that CLIP-Score is a good metric to discriminate between different kinds of visual attributes. We have added a new section in the paper on this (see Sec. (P) - Appendix).
> > In particular, we use the captions in our prompt-dataset to generate images corresponding to different visual attributes.  We then annotate each image based on the visual attribute which was used to generate the image. Then within the category of a given visual attribute, we use CLIP-Score to classify the generated images into the right categories. Overall, we find that CLIP-Score is effectively able to perform well in all the categories of visual-attributes, thereby highlighting its efficacy in our specific use-case.
> > However, we also highlight that there are potential open future directions which can be explored to improve upon CLIP-Score (e.g., using a VQA model for color or a specialized style detector for style) for text-to-image models in general – however that warrants a separate investigation in itself.
> >
> > **The paper highlights the difference in causal state locations between Stable-Diffusion and GPT. Is such a difference specific to the CLIP encoder or is general to the diffusion model class?** : We highlight that we perform causal tracing in two components of the diffusion model : (i) UNet and (ii) Text-Encoder. In the UNet, we find that the causal states are distributed across the whole model, while in GPT causal components are localized within a window of layers. This distribution is primarily due to the diffusion model class, as the UNet is optimized during pre-training. For the text-encoder, we find only one causal state, as opposed to GPT’s multiple causal states within a window. Given that the text-encoder is frozen during pre-training, we believe this surprising observation is more due to CLIP’s pre-training objective. Our general hypothesis is that language models trained with different pre-training objectives will contain causal states in a different way. A full study of such phenomena across different language models is a good direction for future study!

---

> > ### Author Response · Authors · 2023-11-21
> > **Looking Forward To Your Response!**
> >
> > We thank you again for your valuable feedback and comments which has helped to strengthen our paper. As the discussion period is ending soon, we would really appreciate if you could let us know if our responses have addressed your concerns. We will be happy to answer any further questions and address any remaining concerns to the best of our abilities in the remaining time!

---

### Official Review · Reviewer_Qvyq · 2023-11-03

**Soundness:** 4 excellent
**Presentation:** 3 good
**Contribution:** 4 excellent
**Rating:** 8
**Confidence:** 4

**Summary:**

The paper examines the localization of knowledge in language-text models for image generation, specifically UNet and stable diffusion with the CLIP-ViT-L/336px text-encoder. Previous works have not explored these models, and the paper presents some interesting findings such as the wide dispersion of visual concepts across UNet layers coupled with the concentration of text knowledge in a single encoding layer. The latter result is exploited in a new editing algorithm that directly adjusts the weights in the localized component, resulting in high efficiency with similar performance.

The paper is generally clear, the topic is very timely and significant, and the experiments are reasonable.

**Strengths:**

The use of causal mediation analysis is a good idea, and seems to provide a basis for knowledge tracing, but the text should be clearer about how the approach actually uses or follows the CMA paradigm. It is mentioned in the introductory sections but not later on.

The knowledge analysis provides significant insight into multi-modal models, showing they seem to store knowledge differently from language-only models.

Knowledge localization analysis is used effectively to enable highly efficient control and editing of the image generation process, through direct model parameter adjustment without any model training. This seems to be a unique advantage of this approach that is reminiscent of manipulating eigenvectors to generate a range of plausible face images (pre-deep learning).

It seems that the relative importance of each model component is calculated by corrupting it with gaussian noise; generating an image from the corrupted model; then using CLIP-Score to measure the difference between the image and its caption. A low score indicates incompatibility, and implies that the model component is important relative to the caption. This process is linear in the number of model components being tested, and therefore does not scale well to fine-grained components such as individual neurons.

Diff-QuickFix is a clever way to edit generated images, by directly optimizing the weight matrix of a single layer in the text-encoder rather than standard model updating. It is much faster, and appears to be quite effective based on the provided qualitative results.
The experimental design seems sound, comparing against two recent editing methods on the same prompt dataset they used previously, via the CLIP-score metric. The results show that the method achieves equivalent editing performance to the baselines, with a huge gain in computational efficiency.

**Weaknesses:**

The intro is unclear about key points, such as what forms of visual knowledge the paper is focused on, because it is unclear what “visual attribute” means in this paper. In computer vison, an attribute is usually a property of an object such as its color, texture, gender (for a person), presence of accessories (eyeglasses, hats, etc.), age, and so on. It seems that visual attribute here means any sort of visual information, which is confusing.

Fig. 2, which is very effective and interesting, shows that there is a large overlap between model components that are causal for the four different attributes. There are a few components (Unet layers) that are unique to different attributes, but most seem to be causal for all attributes, which is problematic. Not only is information distributed widely across the layers for all attributes, but it is largely the same layers that seem to be encoding most information. This is not a weakness of the causal tracing approach per se, but it does call into question the proposed editing method. How effective could it be, when very few model components independently control different attributes?

Fig. 3 seems highly unlikely and may call the approach into question, as the illustrated result strongly implies that only one layer and one token in that layer encodes relevant knowledge. This seems highly unlikely. Is there some way to corroborate this result?

Most of the references lack information on the venues in which the works were published. This is unusual and improper for a research paper, although it should be easy to fix.

**Questions:**

As above.

---

> ### Author Response · Authors · 2023-11-18
> **Response to Reviewer**
>
> We thank the reviewer for the constructive comments and appreciating our work!
> Below we answer the reviewer's questions:
>
> **The intro is unclear about key points, such as what forms of visual knowledge the paper is focused on, because it is unclear what “visual attribute” means in this paper. In computer vison, an attribute is usually a property of an object such as its color, texture, gender (for a person), presence of accessories (eyeglasses, hats, etc.), age, and so on. It seems that visual attribute here means any sort of visual information, which is confusing** : We thank the reviewer for pointing out that this information is confusing in the introduction! We have modified the introduction accordingly. In the new version of the introduction, we have defined visual attributes as used in our paper. We hope that this new definition in the introduction, makes it less confusing for the readers!
>
> **Fig. 2, which is very effective and interesting, shows that there is a large overlap between model components that are causal for the four different attributes. There are a few components (Unet layers) that are unique to different attributes, but most seem to be causal for all attributes, which is problematic...** : The reviewer raises an interesting question!  It is true that the knowledge corresponding to different visual attributes (e.g., objects / style) is distributed in the UNet. While there are certain layers which are uniquely causal to a particular visual attribute (e.g., self-attn-0-down for style), there are a number of layers which encode common information.
> In the initial part of the project, we wanted to find localized states in the UNet, so that fast closed form edits can be crafted to edit concepts. However, we find that it is indeed not the case, which makes it difficult to perform closed-form edits in the UNet itself. To the best of our knowledge, our work is the first to highlight the difficulty of knowledge editing in the UNet. However, we find that knowledge corresponding to different attributes is localized in the text-encoder – and there exists only one causal state. This unique location which can control the output of the generative model, enables us to design closed-form updates to remove or update information. We also want to highlight that it need not be the case that for effective model editing – few model components independently control different attributes. In general, the primary requirement for effective closed-form model editing is localized causal states – which is satisfied by the text-encoder but not by the UNet. Empirically, we find that even though there is only one causal state (across different attributes) – effective edits can be made as the knowledge is localized.
>
> **Fig. 3 seems highly unlikely and may call the approach into question, as the illustrated result strongly implies that only one layer and one token in that layer encodes relevant knowledge. This seems highly unlikely. Is there some way to corroborate this result?**: We have verified this experiment multiple times during the project! Although this is a surprising observation, the result is indeed true – there exists only one causal state in the CLIP text-encoder across Stable-Diffusion variants – which is the token embedding corresponding to the last subject token in the first self-attention layer. In this figure, we use a threshold on the CLIP-Score such that only the causal states are shown (which is only one corresponding to the last subject token in the first self-attention layer). In Sec. (C) – we have provided more qualitative results showing that there exists only one causal state in the text-encoder. In Sec.P.1, we have provided additional new qualitative results for CLIP ViT-H also, corroborating our current findings.  We also highlight that this information is verified by model editing : In Fig. (4-a), Sec. (5.3) in the main paper and Sec. (G) - Appendix, we essentially discuss the point raised by the reviewer, where we show that using DiffQuickFix on any layer other than the identified causal layer does not lead to any edits.  This result shows and confirms that the identified layer is causally encoding relevant knowledge. We are happy to answer any further questions regarding this experiment!
>
> **Most of the references lack information on the venues in which the works were published. This is unusual and improper for a research paper, although it should be easy to fix.** : We thank the reviewer for bringing this to our notice! We apologize for the error and have fixed the citations in the updated version of our paper!

---

> > ### Author Response · Authors · 2023-11-21
> > **Looking Forward To Your Response!**
> >
> > We thank you again for your valuable feedback and comments which has helped to strengthen our paper. As the discussion period is ending soon, we would really appreciate if you could let us know if our responses have addressed your concerns. We will be happy to answer any further questions and address any remaining concerns to the best of our abilities in the remaining time!

---

### Official Review · Reviewer_TjyF · 2023-11-08

**Soundness:** 2 fair
**Presentation:** 3 good
**Contribution:** 2 fair
**Rating:** 5
**Confidence:** 4

**Summary:**

This paper first introduces the authors' discovery of what are the critical layers for a text to image diffusion model to generate images of certain visual concepts and styles. Based on the analysis, authors propose a method called "DIFF-QUICKFIX" that removes/edits certain visual concepts that the model is able to generate.

The method for localizing these critical layers are adapted from Casual Mediation Analysis, where certain activations of a corrupted model is replaced with that of the original model to identify its impact to the generated image. Specifically both the UNet and text encoder are studied, and experiments show the inspected visual concepts and attributes are scattered on various layers of UNet, as well as the first transformer layer of the CLIP text encoder.

Based on the findings above, authors propose to remove/update concepts by modifying the out put projection unit in the first self attention layer to remap the generated activation to target concept's activation, thus removing/updating the model's generated content.

**Strengths:**

The proposed method for locating and editing knowledge in diffusion models is a novel approach.

The editing method introduced in this paper is unique and significantly faster than traditional training-based techniques. As demonstrated by the results presented in the appendix, the method effectively removes or modifies unwanted concepts introduced by the diffusion model.

The authors clearly convey the paper's main idea, with appropriate background information at most places. The extensive results provided in the appendix are particularly valuable given the lack of well-established metrics for evaluating open-domain text-to-image generation models. These results allow reviewers and readers to effectively assess the method's soundness and effectiveness.

**Weaknesses:**

The scope of the claims seems too broad. The title and introduction claim to locate the knowledge of text-to-image diffusion models, while in the paper, only one stable diffusion model checkpoint is investigated. Given that most of the findings on this model are through laborious experiments, it is unclear if these findings can be generalized to even other versions of stable diffusion models, not to mention other types of text-to-image models. If the findings are only applicable to the studied checkpoint, the impact of the method may be significantly restricted, as the studied model is not considered state-of-the-art.

Lack of comparison to other methods. The visualized results are definitely pleasing to the eye, but it would be better if results of other models could be visualized side-by-side to provide more references.

The subsection on "Selecting Threshold for CLIP-Score" can be confusing to read at the beginning. Perhaps some context of why this is needed would be helpful for readers.

**Questions:**

My main question is how the results of this study generalize to other text-to-image generation models, and what are the components that can be reused if we want to apply the method to another model.

Other than optimizing the W_out, an alternative approach would be to directly replace the token embeddings, e.g., from embeddings for "Van Gogh" to those for "painting". I wonder what the authors' view is on the effectiveness of this approach.

I don't see any results on removing trademarked objects as claimed at the top of page 3. Were they removed during the preparation of the manuscript?

While it seems reasonable to use CLIPScore as AIE, it does not seem that |CLIPScore(x_0^{restored}, c) - CLIPScore(x_{0}^{corr}, c)| can be approximated by CLIPScore(x_0^{restored}, c). In particular, both increases and decreases of CLIPScore(x_0^{restored}, c) by the same amount around CLIPScore(x_{0}^{corr}, c) lead to the same absolute difference, but the CLIPScore(x_0^{restored}, c) is obviously different.

---

> ### Author Response · Authors · 2023-11-18
> **Response to Reviewer - Part 1**
>
> We thank the reviewer for appreciating the novelty of our approach, its effectiveness in concept-ablation and our extensive results!
> Below we answer the reviewer's questions:
>
> **The scope of the claims seems too broad. The title and introduction claim to locate the knowledge of text-to-image diffusion models, while in the paper, only one stable diffusion model checkpoint is investigated. Given that most of the findings on this model are through laborious experiments, it is unclear if these findings can be generalized to even other versions of stable diffusion models...** :  The reviewer raises an interesting question about the generalizability of our framework and results for other checkpoints of Stable-Diffusion.  We first want to highlight that the current version of Stable-Diffusion in our paper is also used in other fine-tuning based concept ablation works (e.g., Concept-Ablation[1], Concept-Erasure[2]).  Therefore to maintain consistency and for fair comparison, we use the current Stable-Diffusion checkpoint. We also note that each component of our paper (causal tracing for UNet and causal tracing for text-encoder) can be independently used to probe any text-to-image model and if the localization property arises, then DiffQuickFix can be used to further edit the model. We have also experimented with our causal tracing + editing framework, in other Stable-Diffusion versions (v1-5 and v2-1). In Sec.(R) and Sec.(Q) - we show that our interpretability and editing method can be used with other SD versions. In fact, we find similar observations in other Stable-Diffusion checkpoints – Knowledge about various visual attributes is localized in the CLIP text-encoder (see Sec. (Q.1)), whereas it is distributed in the UNet (see Sec. (R)). In fact, we show that our editing method works well across the latest Stable-Diffusion checkpoints (see Fig. (60) and Fig. (61) - Appendix).
>
> We have also experimented with other diffusion models such as SD-XL [3](Sec S) and OpenJourney[4] (Sec. R.1.3, R.2). In SD-XL, which is a recent text-to-image model consisting of a much larger UNet (consisting of 2.6B parameters) – we find that causal states are distributed in the UNet like the Stable-Diffusion checkpoints. However for SD-XL we find that only 12 causal layers exist out of 227 layers (~5% of the total layers).  This shows that our interpretability framework can be effectively used to probe other text-to-image generative models. As the reviewer noted that for each of the models, laborious experiments are needed – a full study of SD-XL is attributed for future work. For OpenJourney, we were able to perform successful edits, as shown in Sec. R.2.
>
> In Kandinsky[5] text-to-image generative model (see Sec. (U)) – we find only one localized causal state in the text-encoder, thereby enabling fast edits using DiffQuickFix.
>
> Overall our paper is based on the idea that if there exists localized causal components in some part of the model, then fast edits can be made to remove or edit concepts. Each component of our paper can be used for other text-to-image generative models as shown in the recent experiments. For e.g., the interpretability framework using Causal Mediation Analysis can be used to identify if localized knowledge exists or not. Once localized knowledge is found, then our editing method DiffQuickFix can be used to remove or edit concepts effectively.
>
> [1]. Nupur Kumari, Bingliang Zhang, Sheng-Yu Wang, Eli Shechtman, Richard Zhang, and Jun-Yan Zhu. Ablating concepts in text-to-image diffusion models. In ’International Conference on Computer Vision (ICCV)’, 2023.
>
> [2]. Rohit Gandikota, Joanna Materzynska, Jaden Fiotto-Kaufman, and David Bau.
> Erasing concepts from diffusion models. In Proceedings of the 2023 IEEE International Conference on Computer Vision, 2023.
>
> [3].Podell et al. 2023, arXiv.  https://arxiv.org/pdf/2307.01952.pdf
>
> [4]. OpenJourney:  https://huggingface.co/prompthero/openjourney
>
> [5]. Kandinsky: https://huggingface.co/docs/diffusers/api/pipelines/kandinsky
>
> **Lack of comparison to other methods. The visualized results are definitely pleasing to the eye, but it would be better if results of other models could be ..**:  In Sec.(O) - Appendix, we have provided qualitative results where the generations from different edited models are compared! Given our Appendix has become long, we refrained from providing more qualitative examples, but we will create a project-page with more qualitative visualizations for the final version!

---

> > ### Author Response · Authors · 2023-11-18
> > **Response to Reviewer - Part 2**
> >
> > **The subsection on "Selecting Threshold for CLIP-Score" can be confusing to read at the beginning. Perhaps some context of why this is needed would be helpful for readers** : In our updated version of the paper (see updated Sec. (3.5)), we have provided more context on why a threshold selection mechanism is important for CLIP-Score to select causal states automatically.  To reiterate, we observe that the difference between the CLIP-Score of generated images (after restoring a layer) with high fidelity to the original caption and generated images (after restoring a layer) with low fidelity to the original caption, to be small. Therefore to effectively find a reasonable cut-off point to automatically select causal states (where the generated images have high-fidelity to the original caption), we use our threshold selection mechanism.
> >
> > **Other than optimizing the W_out, an alternative approach would be to directly replace the token embeddings, e.g., from embeddings for "Van Gogh" to those for "painting". I wonder what the authors' view is on the effectiveness of this approach**: The reviewer brings up a good suggestion about alternative mechanisms for model editing, apart from the one we have provided. Intuitively, there are four pathways to go from text-prompt to generated image in a diffusion model. These pathways are : from user's prompt input to text-encoder, from text-encoder to UNet, from UNet to Image-Decoder, and from output of Image-Decoder back to the user. Disrupting any of these pathways (in a multitude of ways) can be effective in concept editing.
> > For example, one could intercept the user's original prompt and replace words or phrases using a look-up table (but a simple look-up table fails to handle mis-spellings or paraphrases [1]). One could similarly run an image classifier on the output image (eg : a "Pikachu detector", where "Pikachu" is a trademarked object owned by Nintendo Inc), and decline to show the user this output. In reality, one might need to combine such interventions to create a truly robust concept-ablation system.  With respect to the reviewer’s suggestion, it is not immediately clear to us how replacing token embeddings would work for concepts with multiple subwords or mis-spellings (we refer reviewer to Section K, which shows our model is robust to mis-spellings, a common real-world attack used to bypass safety features in diffusion models[1]). It might be possible to do this using a look-up table, but vendors would now need to ship this look-up table/ban-list along with the model, whereas our method can just ship the model. In summary, we agree with the reviewer in spirit. An analogy we have often used in internal discussions has been : one can cure poor eyesight using different mechanisms - wearing glasses, wearing contact lenses, performing LASIK surgery, or perhaps even performing neuro-surgery in the brain. Each of these methods has pros and cons, appropriate to the patient's circumstances, ease of technological availability, recovery period, etc. It helps to have these multiple options at our disposal, and combinations of these options may be required at times. We provide one such novel option in our submission.
> >
> > [1] Hongcheng Gao, Hao Zhang, Yinpeng Dong, and Zhijie Deng. Evaluating the robustness of text-toimage
> > diffusion models against real-world attacks, 2023.
> >
> >
> > **I don't see any results on removing trademarked objects as claimed at the top of page 3. Were they removed during the preparation of the manuscript?**:  We refer the reviewer to Section F.2 and Section H. R2-D2 is a robot from star-wars, trademarked by Lucasfilm Ltd LLC. Snoopy the dog is trademarked by Peanuts LLC. Nemo is an animated fish trademarked by Disney Enterprises, Inc. Grumpy cat is trademarked by Grumpy Cat Limited. We show their individual concept ablation in Section F.2 - i.e., four edited models, each unable to generate the trademarked concept that was erased from it. In Section H, we show that all of these concepts can be simultaneously erased by our model, i.e.,the same model is unable to generate all four trademarked objects. We have added this information to the Figures in Section F.2 to add more context to the reader.
> >
> > **While it seems reasonable to use CLIPScore as AIE, it does not seem that |CLIPScore(x_0^{restored}, c) - CLIPScore(x_{0}^{corr}, c)| can be approximated by CLIPScore(x_0^{restored}, c).** : The reviewer raises a valid point. We have modified the approximation (by removing the absolute |.| operation), where now (CLIPScore(x_0^{restored}, c) - CLIPScore(x_{0}^{corr}, c)) can be approximated by CLIPScore(x_0^{restored}, c), as CLIPScore(x_{0}^{corr}, c) is common across all the layer interventions for a given caption c. We have updated this detail in the paper. Note that even with this modified approximation, the results don’t change in our paper, as AIE is still used as CLIPScore(x_0^{restored}, c).

---

> > > ### Author Response · Authors · 2023-11-21
> > > **Looking Forward to Your Response!**
> > >
> > > We thank you again for your valuable feedback and comments which has helped to strengthen our paper. As the discussion period is ending soon, we would really appreciate if you could let us know if our responses have addressed your concerns. We will be happy to answer any further questions and address any remaining concerns to the best of our abilities in the remaining time!

---

> > > > ### Comment · Reviewer_TjyF · 2023-11-22
> > > >
> > > > I appreciate reviewers efforts in addressing these feedbacks.  Several minor questions are clarified in the updated manuscript and author responses.
> > > >
> > > > I would keep my current rating, as my main concern about the over claimed scopes are still there. Through added new experiment results, the authors try to demonstrate that the experiment findings in this paper also apply to other text to image generation models, while in fact these are not. As the authors suggests, the SDXL model's behavior is very different from the V1 series, where the studied concepts are located in only ~5% of the layers. If readers only read this work ( the authors claim findings on SDXL will be in another paper), they will be mislead to believe the concepts are spread over a lot of layers on all text to image diffusion models.
> > > >
> > > > I agree that the proposed model editing method is effective for the proposed tasks, only that the over claimed points make me feel uncomfortable to recommend accepting this paper as is. I am also fine if this impression is only confined to myself, and other reviewers are okay with this approach.

---

> ### Author Response · Authors · 2023-11-22
> **Response to Reviewer**
>
> We thank the reviewer for their response! We want to highlight that our methodology is general but its applications to different text-to-image generative models can reveal their inherent model specific behavior. We have added a new section highlighting these points (see updated Sec. (W)) -- that although our interpretability framework can be used to find knowledge distribution in different text-to-image models, their inference about the distribution ratio about different visual attributes might be different due to different UNet architectures. We believe that this in itself is a novel finding and it's an artifact of the model behaviour which our causal tracing framework is able to articulate.
>
> Although the UNet in SD-XL is almost 3 times bigger than SDv-1.4 and SDv-2.1 -- we still find that knowledge is distributed in the UNet,  but the ratio of distribution is different from Stable-Diffusion versions (as emphasized in Sec. (S)).  For e.g., down-1-resnets.1 is causal as well as mid-blocks.attention.0.transformer-blocks.0.attn1 , up-blocks.0.resnets.2 are causal and these layers are located very far from one another in the Unet. This signifies that knowledge is indeed distributed in the UNet for SD-XL.
>
> We also highlight that our work is more comprehensive than existing published works such as [1,2] which show editing performance on only one Stable-Diffusion variant.  During the rebuttal we have shown the generalizability of our editing framework to multiple Stable-Diffusion checkpoints, OpenJourney and Kandinsky!
>
> [1]. Nupur Kumari, Bingliang Zhang, Sheng-Yu Wang, Eli Shechtman, Richard Zhang, and Jun-Yan Zhu. Ablating concepts in text-to-image diffusion models. In ’International Conference on Computer Vision (ICCV)’, 2023.
>
> [2]. Rohit Gandikota, Joanna Materzynska, Jaden Fiotto-Kaufman, and David Bau. Erasing concepts from diffusion models. In Proceedings of the 2023 IEEE International Conference on Computer Vision, 2023.
>
> We sincerely hope that these additional points in the paper and in this comment, will serve as a pointer to provide a full picture of causal tracing to the reader across various Unet architectures and alleviate the reviewer's concerns.

---

### Author Response · Authors · 2023-11-20
**Global Response to Reviewers**

We thank all the reviewers in their constructive comments towards improving our paper!

We note that all the reviewers appreciated our interpretability analysis for text-to-image models and its further use to design a fast editing method for ablating concepts!

In addition – Reviewer TjyF  found novelty in our proposed approach and our extensive results throughout the paper; Reviewer Qvyq appreciated our elaborate findings and the effectiveness of our editing method - DiffQuickFix; Reviewer cjqU highlighted the empirical advantages that our editing method brings when compared to other methods; Reviewer cqRW appreciated our motivation, findings and extensive experiments.

We have responded individually to the reviewers and have updated the paper based on the comments!  Below we provide a summary of the major updates to the paper:

- Reviewer TjyF, cjqU : We highlight that during the rebuttal period - we have run additional experiments on various open-source text-to-image generative models including Stable-Diffusion Variants (v2-1, v1-5), OpenJourney, Kandinsky and SD-XL. We have provided qualitative and quantitative results in Sec. (Q), Sec. (R), Sec. (S) and Sec. (U).  Overall, our additional results show that our interpretability and editing framework is generalizable to other open-source text-to-image generative models!
- Reviewer TjyF : Updated text in Sec. (3.5) to improve the readability on the selection of a threshold for CLIP-Score.
- Reviewer TjyF: Updated Sec. (F.2) to provide information on the trademarked objects.
- Reviewer Qvyq : Updated Introduction (Sec. 1) to provide better clarity on visual attributes as used in our paper.
- Reviewer Qvyq: The references for a few papers have been fixed in the updated paper.
- Reviewer cjqU: Added a Sec. (P) about the efficacy of CLIP-Scores for the visual-attributes and probe dataset used in our paper.
- Reviewer cqRW: Added a new Sec.(V) which shows that our framework can be used to remove full objects from the scene
- Reviewer cqRW: Fixed the captions in Fig. (6) about the background of a few generated images.


To the best of our knowledge, ours is one of the first papers to explore the inner workings of multimodal generative models and our framework together with its findings can benefit the research community!

---

### Public Comment · ~Yaoteng_Tan1 · 2025-05-09
**Code release for DiffQuickFix**

I appreciate the contributions of this paper and really like it.
I saw that the abstract of the camera-ready version includes a github link for the code ([https://github.com/samyadeepbasu/DiffQuickFix](https://github.com/samyadeepbasu/DiffQuickFix)), but the repository still seems to be empty and hasn’t been updated for over a year. I can see there are a few public requests in the github Issues tab as well. I’d really appreciate it if the authors could consider releasing the code, if possible, as it would be super helpful for the community. Thanks!

---

> ### Public Comment · ~Samyadeep_Basu1 · 2025-05-09
> **Code is available**
>
> The code is open-sourced at : https://github.com/adobe-research/DiffQuickFixRelease.

---

> > ### Public Comment · ~Yaoteng_Tan1 · 2025-05-09
> > **Thank you**
> >
> > Thank you for the updated link, Samyadeep!

---

### Meta-Review · Area_Chair_6wRn · 2023-12-11

**Metareview:**

This paper first studies how stable diffusion model stores and controls the certain visual concepts and styles in its model components, presents the authors’ discovery of what are the critical layers.

Pros (from reviewers):
1. Interesting idea of applying causal mediation analysis on diffusion model for tacking knowledge.
2. The analysis is solid and conclusive, providing good insight of how text-to-image diffusion model stores the knowledge in it network.
3. Introducing a fast image editing method based on the analysis results.

Cons (from reviewers):
1. Over claim the scope of the paper on all text-to-image generation models.
2. Lack of comparison with other methods.

To mitigate the issue of over-claiming, the authors extended their method to more diffusion models, albeit with only limited qualitative results. It is suggested to enhance the paper by providing additional clarification on its scope, reconsidering the title and introduction, or conducting a more in-depth analysis of other diffusion models.

**Justification For Why Not Higher Score:**

See the drawback in the previous section.

**Justification For Why Not Lower Score:**

I believe the absence of baselines and quantitative evaluation for the proposed DiffQuickFix approach might lead to a lower score. However, since none of the reviewers have raised this concern, I am hesitant to override their comments given a average score of 6.5.

---

### Decision · Program_Chairs · 2024-01-16

Accept (poster)